# Safety Assessment and Probiotic Potential of a Novel Species *Lactobacillus xujianguonis*

**DOI:** 10.3390/nu17213474

**Published:** 2025-11-04

**Authors:** Xiaoying Lin, Xiaohui Zhou, Yao Lu, Zheyu Yuan, Ruiting Lan, Ying Du, Liyun Liu, Jianguo Xu

**Affiliations:** 1School of Public Health, Nanjing Medical University, Nanjing 211166, China; 2National Key Laboratory of Intelligent Tracking and Forecasting for Infectious Diseases, National Institute for Communicable Disease Control and Prevention, Chinese Center for Disease Control and Prevention, Beijing 102206, China; 3School of Biotechnology and Biomolecular Sciences, University of New South Wales, Sydney 2052, NSW, Australia; r.lan@unsw.edu.au; 4Hebei Key Laboratory of Intractable Pathogens, Shijiazhuang Center for Disease Control and Prevention, Shijiazhuang 050010, China; 5Department of Epidemiology, School of Public Health, Shanxi Medical University, Taiyuan 030001, China

**Keywords:** *Lactobacillus xujianguonis*, probiotics, safety, oral toxicity

## Abstract

**Background**: Some lactobacilli strains have been documented to cause bacteremia and sepsis in immunocompromised or critically ill hospitalized patients, challenging the universally presumed safety of lactobacilli. Therefore, strain-specific risk assessments are required for the use of *Lactobacillus* as a probiotic. *Lactobacillus xujianguonis*, a novel *Lactobacillus* species isolated from *Marmota himalayana*, has probiotic potential but lacks safety data. **Objective**: To evaluate the preclinical safety of *L. xujianguonis* for food-grade use. **Methods**: Systematic safety assessment includes *in vitro* studies and oral toxicity studies. *In vitro* studies encompassed gastrointestinal tolerance, auto-aggregation and pathogen inhibition, antibiotic susceptibility, and hemolysis/gelatinase activity assays. Oral toxicity studies contained acute single-dose and repeated-dose 28-day oral toxicity studies in mice based on the OECD toxicity study guidelines. **Results**: *L. xujianguonis* strains HT111-2 and 06-2 demonstrated certain probiotic traits, including high acid/bile tolerance, strong auto-aggregation, and antimicrobial activity against common human gastrointestinal pathogens. *In vitro* safety assessments showed susceptibility to nine antibiotics and absence of hemolytic/gelatinase activity. Acute oral exposure (1 × 10^11^ CFU/kg) induced no mortality, clinical abnormalities, or organ toxicity. Subchronic 28-day administration (multiple doses) showed absence of adverse clinical signs with body weight stability and no hematological, biochemical, or histopathological deviations in C57BL/6 mice. Inflammatory and immunological markers remained unaffected. Histological staining results and transcriptional level validation revealed no evidence of intestinal tissue damage. **Conclusions**: This study provides preliminary evidence of the safety of *L. xujianguonis*, supporting its advancement to functional research.

## 1. Introduction

Probiotics, defined as live microorganisms conferring health benefits when administered in adequate amounts, have been extensively utilized in clinical and nutritional applications [1]. Particularly, lactic acid bacteria (LAB) species like *Lactobacillus* demonstrate therapeutic potential across multiple pathological conditions, including gastrointestinal disorders, respiratory infections, metabolic disorders, and immune dysregulation states [2,3]. Their mechanisms encompass gut microbiota modulation through competitive exclusion of pathogens, epithelial barrier reinforcement via tight junction protein stimulation, and systemic immunomodulation [4].

Safety evaluation remains paramount for all consumable products, whether medicinal or nutritional. LAB, designated as Generally Recognized as Safe (GRAS) organisms by regulatory authorities [5], are historically regarded as low-risk microbes due to their status as native gut microbiota and traditional use in fermented foods [6]. However, under certain conditions, lactobacilli may exhibit opportunistic pathogenicity, especially among immunocompromised individuals [7,8]. They can cause serious diseases like *Lactobacillus* bacteremia, endocarditis, hepatic and dental abscesses, and infections associated with knee prosthetics [9,10,11].

Therefore, to gain the status of the Qualified Presumption of Safety (QPS) in accordance with the requirements of European Food Safety Authority (EFSA), new strains, especially those from non-human origins, require strain-specific evaluation of their efficacy and safety to establish if they are as safe as conventional food-grade organisms and have potential applications in the food or medicinal sectors [12,13]. Under frameworks of the EFSA, thirty-six *Lactobacillus* strains have received QPS designation [13]. The World Health Organization (WHO) and the US Food and Drug Administration (FDA) have developed guidelines for the evaluation of probiotics in food, mandating preclinical and clinical evaluations to validate probiotic safety profiles [1]. These require systematic interrogation of strain-specific risks encompassing antibiotic resistance patterns, toxigenic potential, hemolytic activity, and adverse events in controlled human trials [14].

*Lactobacillus xujianguonis* is a novel species of *Lactobacillus* isolated from the intestinal tract of *Marmota himalayana* on the Qinghai–Tibet Plateau by our group [15]. Its origin is different from that of traditional human-derived or food-derived lactobacilli. Given the uniqueness of its source, this study aims to systematically evaluate the phenotypic characteristics and *in vivo* safety of two representative strains of *L. xujianguonis* in order to investigate its potential as a probiotic. The comprehensive evaluation of the strains encompassed multiple aspects, starting with fundamental biological function characterization to evaluate its survival potential in the gastrointestinal tract, followed by *in vitro* analysis of antibiotic susceptibility and hemolytic or gelatinase activities, and *in vivo* toxicological evaluation including single-dose acute oral toxicity and repeated-dose 28-day oral toxicity studies in C57BL/6 mice.

## 2. Materials and Methods

### 2.1. Bacterial Strains and Culture Condition

*L. xujianguonis* strains HT111-2 and 06-2 were deposited in the China General Microbiological Culture Collection Center (CGMCC) with preservation number CGMCC 1.13855 and CGMCC 23437. Indicator bacteria included *Streptococcus pneumoniae* ATCC 49619, *Staphylococcus aureus* ATCC 25923, *Lacticaseibacillus rhamnosus* GG ATCC 53103, Enterohemorrhagic *Escherichia coli* (EHEC) ATCC 43895, *Salmonella* Typhimurium ATCC 14028, *Listeria monocytogenes* EGD-e ATCC BAA-679, Enterohemorrhagic *Escherichia coli* (EAEC) CICC 24186, and *Shigella flexneri* 2a str. 301. Except for *S. flexneri* 2a str. 301 stored at the National Key Laboratory of Intelligent Tracking and Forecasting for Infectious Diseases (Beijing, China), all other strains were procured from the American Type Strain Preservation Center (ATCC). For revival from frozen storage, *L. xujianguonis* strains and *Lactobacillus rhamnosus* GG (LGG) were cultured anaerobically on de Man, Rogosa Sharpe (MRS) agar (OXOID, Thermo Fisher Scientific, Waltham, MA, USA) supplemented with 5% sheep blood at 37 °C for 48 h. Other reference strains were cultivated aerobically on brain heart infusion (BHI) agar (OXOID, Thermo Fisher Scientific, Waltham, MA, USA) at 37 °C for 24 h.

### 2.2. Phenotypic Safety Assessment and Probiotic Characteristics

#### 2.2.1. Survivability in Low pH or 0.3% Bile Salt

The acid and bile acid tolerance of *L. xujianguonis* strains HT111-2 and 06-2 was evaluated via modified viable cell counting [16]. Briefly, cell suspensions (1 × 10^8^ CFU/mL in phosphate-buffered saline [PBS, pH 7.4; Gibco, Thermo Fisher Scientific, Waltham, MA, USA]) were anaerobically incubated in MRS broth under two stress conditions with three parallel experiments: (1) pH 3.0 for 3 h or (2) 0.3% (*w*/*v*) bile salts for 4 h at 37 °C. Parallel control experiments used neutral MRS broth (pH 7.0) without bile supplementation.$$\mathrm{Survival}\;\mathrm{rate}\;(\%)=\frac{\mathrm{lg}{(CFUN}_{1})}{\mathrm{lg}{(CFUN}_{0})}\times 100.$$

N_1_ represents the total number of viable bacteria exposed to assay conditions and N_0_ represents the number of initially viable bacteria before exposure to assay conditions.

#### 2.2.2. Auto-Aggregation and Cell Surface Hydrophobicity

Auto-aggregation capacity and cell surface hydrophobicity of *L. xujianguonis* strains HT111-2 and 06-2 were analyzed using a protocol adapted from Zulkhairi et al. [17]. For auto-aggregation, bacterial suspensions in PBS were standardized to an initial optical density (OD, A = 600 nm) of 1.0 (OD_0_). After 24 h incubation at 37 °C in an anaerobic chamber, the OD value was measured as OD_1_. Surface hydrophobicity was quantified via xylene adhesion: cell suspensions (OD_600_ = 1.0 in 0.85% NaCl) were combined with equal volumes of xylene, vortexed (2 min), and equilibrated at 37 °C for 30 min. After phase separation, the aqueous layer OD_600_ (OD_1_) was recorded. Three technical replicates were performed in both experiments, with LGG serving as the positive control strain. Both metrics were calculated as [(OD_0_ − OD_1_)/OD_0_] × 100 and categorized per established thresholds: low (0–34%), moderate (35–69%), or high hydrophobicity (70–100%) [18].

#### 2.2.3. Antimicrobial Activity

The antimicrobial activity of *L. xujianguonis* strains was evaluated using an Oxford plate assay as reported previously [19]. Overnight cultures of both strains were inoculated (1% *v*/*v*) into MRS broth and incubated anaerobically at 37 °C for 48 h. Autoclaved Oxford cups (inner diameter: 6 mm, outer diameter: 8 mm, height: 10 mm; Shanghai Precision Instrument Co., Ltd., Shanghai, China) were positioned on solidified MRS agar plates. Subsequently, 15 mL of molten MRS agar (50 °C) was layered over the cups, which were removed after solidification to form uniform wells. Target pathogens (EHEC ATCC 43895, *S.* Typhimurium ATCC 14028, *S. aureus* ATCC 26072, *L. monocytogenes* EGD-e ATCC BAA-679, and *S. flexneri* 2a str. 301) were streaked onto BHI agar at 1 × 10^6^ CFU/mL. Wells received 200 μL aliquots of either sterile MRS broth (negative control) or *L. xujianguonis* fermentation broth. After 12 h of incubation at 37 °C, inhibition zones were quantified. The inhibition zone diameter is measured as the total diameter of the clear, circular area devoid of bacterial growth, excluding the Oxford cup.

#### 2.2.4. Antimicrobial Susceptibility

According to the Performance Standards for Antimicrobial Susceptibility Testing made by the Clinical and Laboratory Standards Institute (CLSI) M45, the antibiotic susceptibility of *L. xujianguonis* was assessed by E-test strips method to determine the minimum inhibitory concentration of penicillin, ampicillin, imipenem, meropenem, vancomycin, daptomycin, erythromycin, clindamycin, and linezolid. *S. pneumoniae* ATCC 49619 was served as quality control.

#### 2.2.5. Hemolysin and Gelatinase Production

Hemolytic and gelatinase activities of *L. xujianguonis* strains were assessed using established protocols [20,21] with minor modifications. For hemolysis evaluation, strains cultured anaerobically on MRS agar supplemented with 5% (*v*/*v*) sheep blood at 37 °C for 48 h were examined for colony-adjacent zones: clear zones (β-hemolysis), greenish zones (α-hemolysis), or no hemolysis (γ-hemolysis). For gelatinase production, 10 μL aliquots of fresh cultures were spot-inoculated onto MRS agar containing 3% (*w*/*v*) gelatin. Following 72 h anaerobic incubation at 37 °C, plates were flooded with saturated ammonium sulfate solution; opaque halos surrounding colonies indicated positive gelatinase activity. *S. aureus* ATCC 25923 served as the positive control.

### 2.3. In Vivo Toxicology Studies

#### 2.3.1. Animal

Oral toxicity studies were conducted using C57BL/6 mice (6–8weeks; Beijing Charles River Laboratory Animal Technology Co., Ltd., Beijing, China) under protocols approved by the Welfare & Ethical Inspection in Animal Experimentation Committee of the Chinese Center for Disease Control and Prevention (Appl. no.: 2025-024). Mice were housed in specific-pathogen-free (SPF) conditions with a 12 h light/dark cycle, ambient temperature of 25 ± 2 °C, and *ad libitum* access to food (following the GB/T 34240-2017 standard [22]) and water. A 7-day acclimation period preceded experimental procedures.

#### 2.3.2. Acute Oral Toxicity Study

Acute oral toxicity was evaluated following the Organisation for Economic Co-operation and Development (OECD) guideline No. 423 with certain modifications from Ou et al. [23]. Eighteen female C57BL/6 mice (n = 6) were randomly allocated to three groups using a random number table: (1) negative control (0.2 mL PBS), (2) *L. xujianguonis* HT111-2 (1 × 10^11^ CFU/kg, 0.2 mL), and (3) *L. xujianguonis* 06-2 (1 × 10^11^ CFU/kg, 0.2 mL) groups. Bacterial suspension or PBS was administered via single oral gavage. Clinical monitoring (lethargy, distress, mortality) occurred at 0.5, 1, and 2 h post-dosing, then daily for 14 consecutive days. Body weights were recorded on days 0 (baseline), 1, 3, 7, and 14. Following the observation period, all mice underwent terminal anesthesia with isoflurane overdose (5% *v*/*v*) and systematic necropsy for gross pathological assessment.

#### 2.3.3. 28-Day Repeated-Dose Toxicity Study

A 28-day subacute toxicity study of *L. xujianguonis* strains was conducted in accordance with OECD Guideline 407 to establish dose–response relationships and determine the no-observed-adverse-effect level (NOAEL). Via random number table method, eighty-four C57BL/6 mice (seven groups of twelve mice each, 1:1 gender ratio) were divided into a negative control (NC) group (PBS, 10 mL/kg/day) and six treatment groups: *L. xujianguonis* HT111-2 or 06-2 administered via daily oral gavage at high (1 × 10^11^ CFU/kg/day, denoted as H1 and H2 group, respectively), medium (2.5 × 10^10^ CFU/kg/day, M1/M2 group), or low (5 × 10^9^ CFU/kg/day, L1/L2 group) doses for 28 consecutive days where 1 and 2 denotes HT111-2 and 06-2 treatment respectively. Daily clinical monitoring included mortality, fur/skin integrity, respiratory patterns, and behavioral changes, with weekly recordings of body weight, food intake, and water consumption. On day 29, mice underwent terminal blood collection for hematological, biochemical, and cytokine analyses, followed by necropsy with gross pathology evaluation, organ/terminal body weight ratio (organ index), histopathological examination, and RT-qPCR analysis of colon tissues for intestinal barrier-related gene expression.

#### 2.3.4. Hematology and Biochemical Testing

Following blood collection, samples were divided into two aliquots: one portion transferred to EDTA-coated anticoagulant tubes (Solarbio, Beijing, China) for complete blood count (CBC) analysis, and the remainder placed in sterile 1.5 mL microcentrifuge tubes (Eppendorf, Hamburg, Germany) for serum separation. Serum was obtained by centrifugation at 3000× *g* for 15 min at 4 °C. CBC parameters were quantified using a Mindray BC-2800Vet automated hematology analyzer (Mindray Bio-Medical Electronics Co., Ltd., Shenzhen, China), while serum biochemical analyses were performed with a Hitachi 3500 automated biochemical analyzer (Hitachi High-Tech Corp., Tokyo, Japan).

#### 2.3.5. Histopathology Analysis

The six vital tissues (liver, kidneys, spleen, heart, lungs, colon) of mice were fixed in 4% paraformaldehyde (Biosharp, Beijing, China) and stained with hematoxylin and eosin (H&E) by Wuhan Servicebio Technology Co., Ltd. (Wuhan, China) according to previous studies [24].

#### 2.3.6. Quantitative Real-Time Polymerase Chain Reaction

Colon segments preserved in RNAlater^®^ Stabilization Solution (Invitrogen, Carlsbad, CA, USA) were homogenized in TRIzol™ Reagent (Ambion, Carlsbad, CA, USA) for total RNA extraction. RNA quality was determined by the A260/A280 ratio, with a value between 1.8 and 2.1 considered acceptable. RNA was reverse-transcribed into cDNA using the PrimeScript™ RT Reagent Kit (Takara Bio, Shiga, Japan) and amplified via RT-qPCR with the following primers: *Gapdh* (forward: 5′-GCAAGTTCAACGGCACAG-3′, reverse: 5′-CGCCAGTAGACTCCACGAC-3′), *Occludin* (forward: 5′-GCCCAGGCTTCTGGATCTATGT-3′, reverse: 5′-GGGGATCAACCACACAGTAGTGA-3′), *Zo1* (forward: 5′-CCTAAGACCTGTAACCATCT-3′, reverse: 5′-CTGATAGATATCTGGCTCCT-3′), and *Muc2* (forward: 5′-GCTGACGAGTGGTTGGTGAATG-3′, reverse: 5′-GATGAGGTGGCAGACAGGAGAC-3′). The 2^−ΔΔCT^ method was employed to determine relative expression levels, with *Gapdh* serving as the reference gene.

### 2.4. Statistical Analysis

Statistical analysis was performed using GraphPad Prism v9.0. The Shapiro–Wilk and F tests were used to evaluate data normality and variance homogeneity, respectively. The experimental data were compared using one-way ANOVA with Tukey’s multiple comparisons test. Spearman correlation analysis was conducted to determine the repeatability of samples and quality control samples in the group. *p* < 0.05 denotes statistical significance.

## 3. Results

### 3.1. Probiotic Properties

The two *L. xujianguonis* strains HT111-2 and 06-2 exhibited comparable survival rates to the commercial probiotic LGG under low pH conditions, with HT111-2 displaying higher viability relative to LGG at 0.3% (*w*/*v*) to bile salt exposure (*p* < 0.01, Figure 1A). Both *L. xujianguonis* strains showed significantly stronger auto-aggregation capacity than LGG (*p* < 0.05) and maintained high hydrophobicity (>70%), a level comparable to that of LGG (Figure 1B). Moreover, both strains exhibited broad-spectrum antimicrobial activity against enteric pathogens, with particularly potent inhibition observed against *L. monocytogenes*, and the two strains have similar efficacy (Table 1).

### 3.2. Safety Assessment In Vitro

Both *L. xujianguonis* strains were susceptible to the *Lactobacillus* panel of nine antibiotics (penicillin, ampicillin, imipenem, meropenem, vancomycin, daptomycin, erythromycin, and clindamycin) (Appendix A) and lacked detectable hemolytic or gelatinase activities.

### 3.3. Acute Toxicity Study

Mice were orally administered the *L. xujianguonis* strains, HT111-2 and 06-2 seperately, to test acute toxicity with a single dose of 1 × 10^11^ CFU/kg body weight. The two treated groups maintained normal vitality throughout the study period, with no mortality, morbidity, or behavioral anomalies observed. Feeding parameters (average daily food and water intake, Appendix A) and body weights (Table 2, *p* > 0.05) showed parity between control and treated groups. Organ weight analysis (heart, liver, spleen, lungs, kidneys) and macroscopic examination revealed no treatment-related abnormalities (Figure 2A–E), with colon morphology remaining comparable across groups (Figure 2F, *p* > 0.05). Gross necropsy detected no pathological lesions; according to OECD 423 guidelines, further histopathological examination was not performed given the absence of clinical signs.

### 3.4. Twenty-Eight-Day Repeated-Dose Oral Toxicity Study

To determine dose-dependent repeat exposure toxicity and any gender difference, female and male mice were separately treated with three different doses of the two *L. xujianguonis* strains [high dose (1 × 10^11^ CFU/kg body weight), medium (2.5 × 10^10^ CFU/kg body weight), and low (5 × 10^9^ CFU/kg body weight)] for 28 days. No mortality, treatment-associated morbidity, or behavioral anomalies occurred across high-, medium-, and low-dose groups and control groups during the 28-day exposure. In both genders, the trend of body weight gain was similar among all groups, and food consumption patterns showed comparable progression across groups (*p* > 0.05; Figure 3, Appendix A). Pathological, immunological, and intestinal parameters were assessed as detailed in sections below.

#### 3.4.1. Clinical Pathology and Immunological Effects

To determine the levels of systemic inflammation, metabolism, and immune responses in mice following different doses of *L. xujianguonis* intervention, we assessed a range of biochemical and immunological markers. Hematological parameters (white blood cell, lymphocyte, monocyte, granulocyte, and red blood cell count, hemoglobin, hematocrit, mean corpuscular volume, platelet count, mean platelet volume, and platelet distribution width) in male and female mice did not differ significantly among the groups, with all parameters within standard reference ranges (Table 3). Biochemical indices (alanine aminotransferase, aspartate aminotransferase, blood urea nitrogen, creatinine, total cholesterol, triglycerides, creatine kinase, sodium, potassium) showed temporal alignment with control values at the measured intervals (Table 4). No treatment-related alterations were detected in serum inflammatory markers (TNF-α, IL-6, IL-1β; Figure 4A–C) or immune modulators (IFN-γ, IL-4, IL-12; Figure 4D–F), maintaining parity with negative controls following 28-day exposure.

#### 3.4.2. Organ Weight, Macroscopic Observations and Histopathology

The organ indices of each group of mice were measured to evaluate the metabolic toxicity of *L. xujianguonis* HT111-2 and 06-2 at different doses. The results showed no significant variation between the control group and treatment groups (Figure 5A,B), and test-item-related changes in microscopic examinations were not observed in either gender in any of the treated groups. Then, we selected six important organs of the control and high-dosing groups of two strains for further pathological analysis. No obvious pathological changes were observed in the organs of the two high-dose treatment groups (H1 and H2 groups; Figure 5C). Among them, hepatic tissue displayed intact lobular architecture with well-defined hepatocytes exhibiting uniform cytoplasmic staining. Intestinal morphology maintained epithelial integrity, showing orderly glandular organization without lymphocytic infiltration or connective tissue disruption (Figure 5C).

#### 3.4.3. Gut Barrier and Translocation Safety

Except for the increased expression of *occludin* in *L. xujianguonis* 06-2 high-dose female and *L. xujianguonis* HT111-2 medium-dose male mice, the gene expression profiles of epithelial barrier markers (*Occludin*, *Zo1*, *Muc2*) in different dose intervention groups of the two *L. xujianguonis* strains in male and female mice were comparable to those in the NC group (Figure 6), which was consistent with the results of systemic pathological indicators (clinical signs, hematology, histopathology). This transcriptional evidence aligns with histological confirmation of preserved mucosal architecture, refuting bacterial translocation risks under experimental conditions.

## 4. Discussion

A strain-specific risk assessment is necessary for the use of *Lactobacillus* as a probiotic [25]. This study systematically investigated the probiotic properties and safety profile of *L. xujianguonis* through integrated *in vitro* assays and *in vivo* models. The result showed that *L. xujianguonis* HT111-2 and 06-2 have promising probiotic properties, such as stress resistance, auto-aggregation, and antibacterial activity. Their safety was supported by antibiotic sensitivity, lack of hemolytic/gelatinase activity, and absence of treatment-related adverse effects in both single-dose and 28-day oral toxicity studies at doses up to 1 × 10^11^ CFU/kg.

For successful gastrointestinal colonization, probiotics must exhibit adherence to intestinal epithelial cells and mucosal surfaces, a prerequisite for evading rapid elimination by intestinal peristalsis [26]. The adhesive capacity and epithelial colonization traits of *Lactobacillus* strains constitute the primary mechanism underlying their competitive exclusion of enteropathogens and subsequent impairment of mucosal adhesion [27]. Meanwhile, the capacity to withstand gastric acidity and bile salt stress is also an essential prerequisite for microbial survival in the gut environment [26]. Existing research indicates that bacterial auto-aggregation capacity correlates strongly with cellular adherence properties [28], with cell surface hydrophobicity emerging as a critical determinant of adhesion efficacy [29]. Notably, our experimental findings demonstrate that *L. xujianguonis* strains HT111-2 and 06-2 display acid and bile tolerance, colonization capacity, and antibacterial activity, comparable to that of LGG. However, the outcomes of *in vitro* assays may not fully predict the colonization potential of probiotics *in vivo*.

The assessment of antimicrobial resistance constitutes critical biosafety consideration in the selection of probiotic therapeutic strains [30]. The QPS framework mandates rigorous evaluation of intrinsic resistance determinants and horizontal gene transfer potential in microbial additives [31]. Both strains evaluated in this study demonstrated full susceptibility to a CLSI-recommended panel of nine antimicrobial agents across multiple drug classes. Genome-based screening will still be needed to determine whether any other resistance properties not identified through phenotypic tests of the limited number of agents.

Acute oral toxicity evaluation has been advocated as a fundamental toxicological assessment for probiotic safety validation, having been extensively adopted in regulatory toxicology research [32]. In mammalian toxicological paradigms, systematic monitoring of feed conversion efficiency, body weight, and organ index provides validated toxicological biomarkers that quantifiably assess substance-induced pathophysiological alterations [33]. Our standardized 14-day acute exposure mouse experiments revealed no statistically significant deviations in these core parameters between experimental and control groups. Crucially, macroscopic pathological analysis demonstrated the absence of strain-associated mortality or tissue-level abnormalities following a single-dose administration of 1 × 10^11^ CFU/kg observed over 14 days. These collective findings establish that the estimated median lethal dose (LD_50_) for two *L. xujianguonis* strains exceeds current maximum tested concentrations, confirming their biosafety profile under acute exposure conditions. Our findings aligned with the changes described in other equivalent oral repeated toxicity studies of LAB, which have shown no toxicological effect in rodent models at daily doses of 10^8^–10^10^ CFU on experimental animals according to acute or subchronic toxicity studies [34,35].

In toxicological characterization of chemical substances, repeated-dose oral toxicity testing should be conducted subsequent to acquiring preliminary toxicity data through acute exposure studies [36]. The assessment characterizes dose-dependent health implications of repeated short-term exposures, notably perturbations in neural regulation, immune competence, and hormonal homeostasis [36]. Our study revealed no strain-related alterations in comprehensive toxicological endpoints, including clinical observations, terminal body weight, hematological profiles, clinical biochemistry parameters, organ indices, or histopathological characteristics.

Notably, hematological analysis provides sensitive indicators for systemic inflammation detection, while serum biochemistry evaluation of hepatorenal functional indices serves as a critical diagnostic triad for assessing metabolic toxicity [37,38]. Previous studies have revealed no significant differences in blood parameters between control groups and those receiving *Lactobacillus plantarum* or *Lactobacillus casei* supplementation [39,40]. Another study reported that hematological parameters, such as hemoglobin level and hematocrit, were comparable to the normal control group in mice after *Kluyveromyces marxianus* treatment [14]. In the present study, hematological parameters were confirmed to be within the normal range, demonstrating that hematological homeostasis was maintained following probiotic intervention. Additionally, integrated assessment of hepatorenal function demonstrated preserved hepatocellular membrane integrity through ALT and AST levels within mouse normal values, alongside maintenance of glomerular filtration capacity evidenced by physiological creatinine concentrations. Normal TC and TG levels, along with cardiac biomarkers (CK, Na, K) within normal limits, confirm stable metabolic function and uncompromised cardiac metabolic homeostasis.

Similarly, cytokines are indicators of the activation of the immune defenses and reactivity of the body to external threats [41]. Proinflammatory cytokines (TNF-α, IL-1β, IL-6) cause acute and chronic inflammation, enhancing various defense mechanisms, especially immunological responses [41]. Changes in immune factors (IFN-γ, IL-4, IL-12) confirm immunomodulatory properties of strain treatment [42]. In this study, cytokine profiling (TNF-α, IL-1β, IL-6, IFN-γ, IL-4, IL-12) showed no significant variations between treatment groups and controls, which confirmed the absence of systemic immunostimulation by the *L. xujianguonis* strains. However, cytokine analysis has limitations, such as a short observation window, and should be viewed objectively.

The intestinal mucosal barrier maintains immunological barrier integrity through strict regulation of pathobiont containment and xenobiotic compound sequestration [43]. Microbial translocation across this barrier represents a critical initiating event in hematogenous dissemination that may progress to bacteremia [44]. Consequently, the microbial translocation potential of *L. xujianguonis* HT111-2 and 06-2 were assessed via an indirect approach, which revealed that mRNA levels of intestinal barrier-related regulatory markers (*Occludin*, *Zo1*, *Muc2*) maintained or exceeded baseline expression levels compared to controls, suggesting that mucosal barrier integrity was likely maintained.

Based on these observations and analyses, we conclude that *L. xujianguonis* HT111-2 and 06-2 administration induced no toxicologically significant alterations in the evaluated spectrum of biomarkers, including physical status indices, hematological profiles, serum biochemistry parameters, or histomorphological architecture across all tested dosages and both genders of mice. The derived NOAEL is 1 × 10^11^ CFU/kg bodyweight/day (2 × 10^9^ CFU/mouse). This may be extrapolated to the human-equivalent NOAEL dose with caution, considering interspecies differences and other factors [32]. Further studies will be required on (1) long-term exposure beyond 90 days, (2) other cross species models for pharmacological safety, and (3) immunocompromised hosts to ensure no increased risk to this population group.

## 5. Conclusions

This study demonstrated that *L. xujianguonis* strains HT111-2 and 06-2 exhibit beneficial potential, including high acid/bile tolerance, strong self-aggregation, and antibacterial activity against common human gastrointestinal pathogens. They also demonstrate certain safety profiles, including sensitivity to nine selected antibiotics, absence of hemolytic/gelatinase activity, and single-dose and 28-day repeated oral safety at a dose of at least 1 × 10^11^ CFU/kg, which provided the basis for further functional research based on probiotic safety [45,46].

## Figures and Tables

**Figure 1 nutrients-17-03474-f001:**
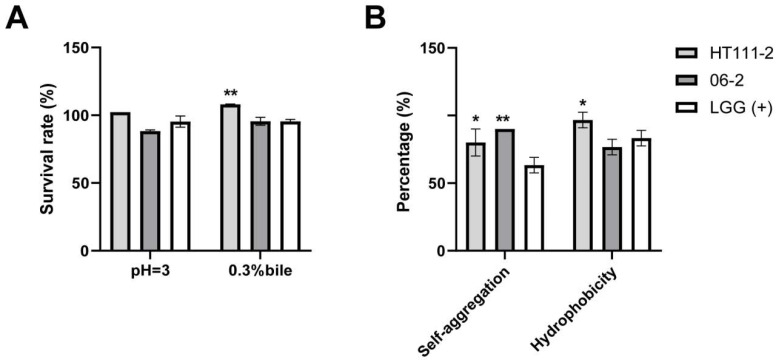
Probiotic properties of *L. xujianguonis* strains HT111-2 and 06-2. (**A**) Survival ability in gastrointestinal tract, including acid and 0.3% bile salt tolerance. (**B**) Colonization capacity in the intestine. Values represent mean ± standard error; LGG: *Lactobacillus rhamnosus*. *: *p* < 0.05, **: *p* < 0.01. number of biological replicates = 3.

**Figure 2 nutrients-17-03474-f002:**
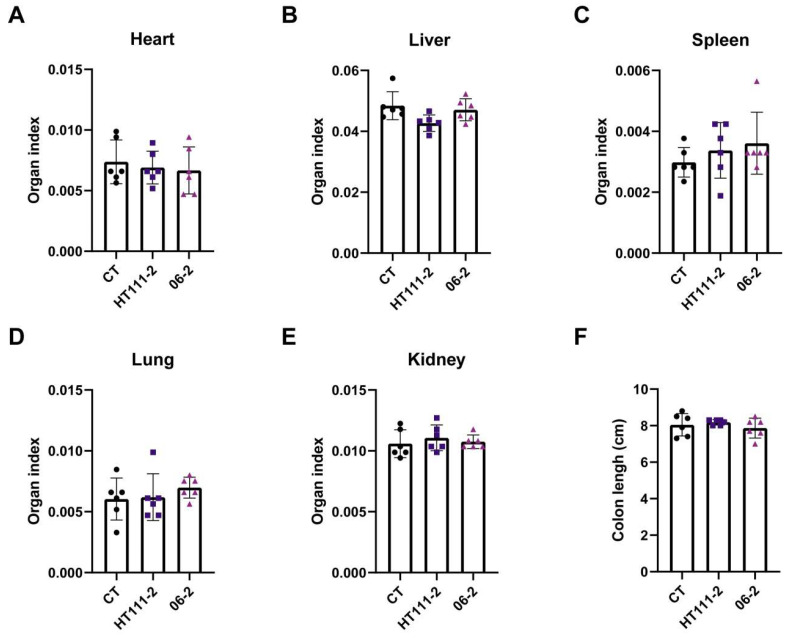
Effects of *L. xujianguonis* strains on organs in the acute toxicity study. (**A**–**E**) Organ index of heart, liver, spleen, lung, and kidney; (**F**) Colon length of mice; Each of the solid round, square and triangle symbols represent a mouse in treatment groups, CT, HT111-2 and 06-2 respectively. Number of mice per group = 6.

**Figure 3 nutrients-17-03474-f003:**
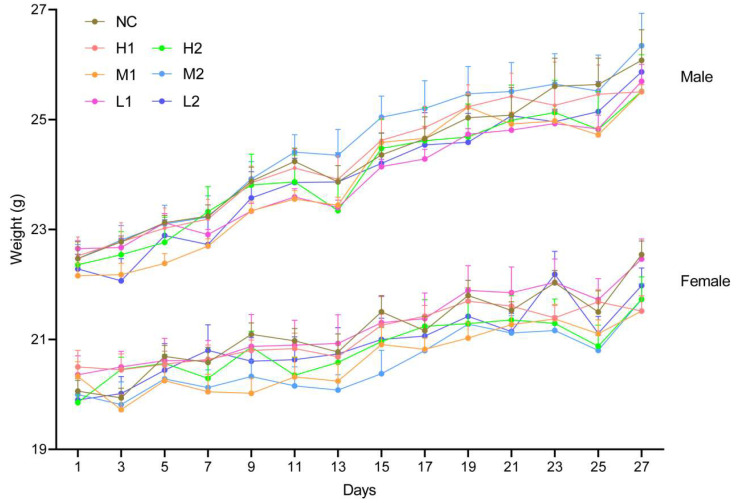
Body weight of mice in the subacute toxicity study. Number of mice per group = 6 for each gender. NC: negative control, H1: high-dose *L. xujianguonis* HT111-2 gavage, H2: high-dose *L. xujianguonis* 06-2 gavage, M1: middle-dose *L. xujianguonis* HT111-2 gavage, M2: middle-dose *L. xujianguonis* 06-2 gavage, L1: low-dose *L. xujianguonis* HT111-2 gavage, L2: low-dose *L. xujianguonis* 06-2 gavage.

**Figure 4 nutrients-17-03474-f004:**
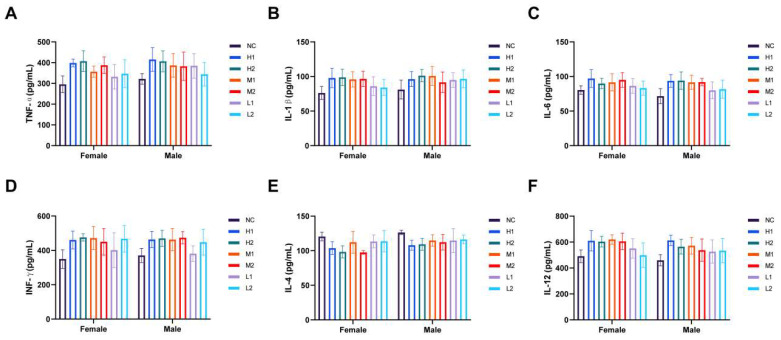
Effects on inflammatory and immune factors after *L. xujianguonis* intervention in the subacute toxicity study. (**A**–**F**) Levels of cytokines, TNF-α, IL-1β, IL-6, INF-γ, IL-4, and IL-12, in mouse serum. NC: negative control, H1: high-dose *L. xujianguonis* HT111-2 gavage, H2: high-dose *L. xujianguonis* 06-2 gavage, M1: middle-dose *L. xujianguonis* HT111-2 gavage, M2: middle-dose *L. xujianguonis* 06-2 gavage, L1: low-dose *L. xujianguonis* HT111-2 gavage, L2: low-dose *L. xujianguonis* 06-2 gavage. number of mice per group = 6 for each gender.

**Figure 5 nutrients-17-03474-f005:**
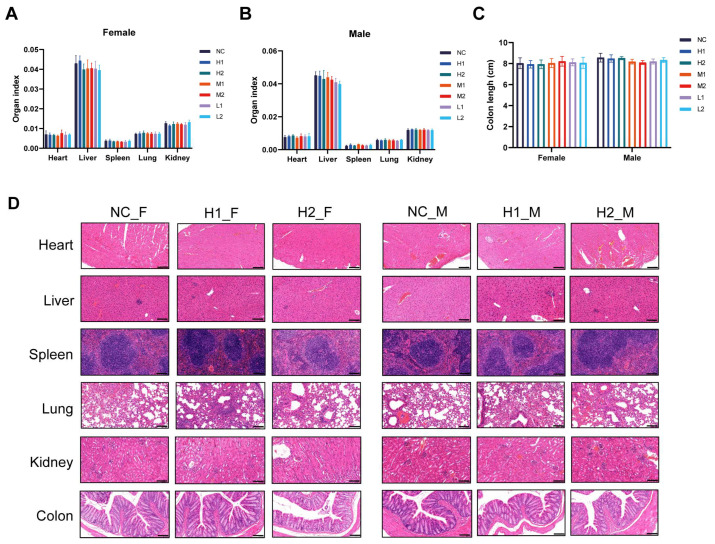
Effects of *L. xujianguonis* strains on organs in the subacute toxicity study. (**A**,**B**) organ index; (**C**) colon length; (**D**) representative images of hematoxylin-eosin (HE) stained histology sections. Bar: 100 µm. NC: negative control, H1: high-dose *L. xujianguonis* HT111-2 gavage, H2: high-dose *L. xujianguonis* 06-2 gavage, M1: middle-dose *L. xujianguonis* HT111-2 gavage, M2: middle-dose *L. xujianguonis* 06-2 gavage, L1: low-dose *L. xujianguonis* HT111-2 gavage, L2: low-dose *L. xujianguonis* 06-2 gavage. In NC_F, NC_M, H1_F, H1_M, H2_F, and H2_M, F and M denote female and male mice respectively. Number of mice per group = 6 for each gender.

**Figure 6 nutrients-17-03474-f006:**
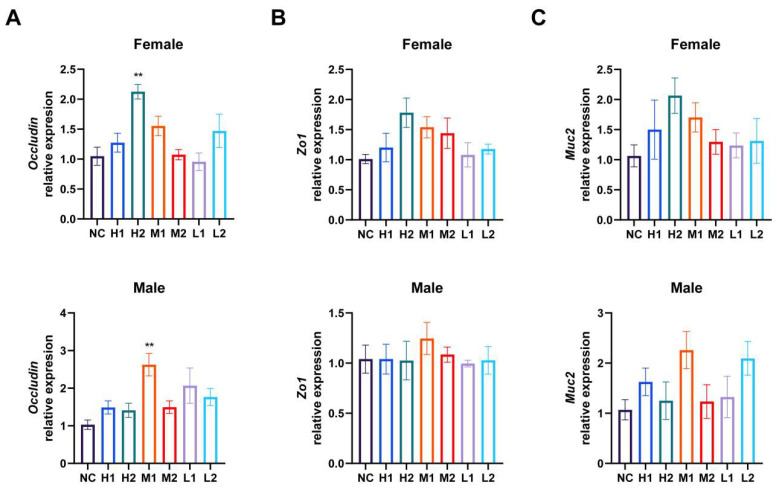
Relative mRNA expression of genes associated with intestinal epithelial barrier functions, including *Occludin* (**A**), *Zo1* (**B**), *Muc2* (**C**). **: *p* < 0.01. NC: negative control, H1: high-dose *L. xujianguonis* HT111-2 gavage, H2: high-dose *L. xujianguonis* 06-2 gavage, M1: middle-dose *L. xujianguonis* HT111-2 gavage, M2: middle-dose *L. xujianguonis* 06-2 gavage, L1: low-dose *L. xujianguonis* HT111-2 gavage, L2: low-dose *L. xujianguonis* 06-2 gavage. Number of mice per group = 6 for each gender.

**Table 1 nutrients-17-03474-t001:** The antimicrobial effects of *L. xujianguonis* strains HT111-2 and 06-2 against six human pathogens.

Pathogenic Species	Inhibition Zone (mm)	
	HT111-2	06-2
EAEC (CICC 24186)	3.83 ± 1.34	3.90 ± 1.21
EHEC (ATCC 43895)	3.50 ± 0.40	3.17 ± 0.67
*S.* Typhimurium (ATCC 14028)	3.03 ± 1.17	3.47 ± 0.65
*S. aureus* (ATCC 26072)	2.53 ± 0.12	3.30 ± 1.77
*L. monocytogenes* EGD-e (ATCC BAA-679)	14.47 ± 2.15	15.20 ± 1.41
*S. flexneri* 2a str. 301	5.36 ± 0.67	3.33 ± 0.95

Abbreviations. EAEC: Enterohemorrhagic *Escherichia coli*, EHEC: Enterohemorrhagic *Escherichia coli*. n = 3.

**Table 2 nutrients-17-03474-t002:** Body weights of mice of different treatment groups in the acute toxicity study ^#^.

Day	CT	HT111-2	06-2
D0	18.14 ± 1.25	18.10 ± 0.49	18.06 ± 0.51
D1	18.44 ± 1.18	18.10 ± 0.45	18.05 ± 0.31
D3	18.60 ± 1.09	18.38 ± 0.70	18.26 ± 0.65
D7	19.34 ± 1.34	19.06 ± 0.92	18.85 ± 0.50
D14	19.75 ± 1.03	19.07 ± 0.91	19.20 ± 0.74

^#^ CT: negative control, HT111-2: C57BL/6 mice treated with *L. xujianguonis* HT111-2, 06-2: C57BL/6 mice treated with *L. xujianguonis* 06-2. Number of mice per group = 6.

**Table 3 nutrients-17-03474-t003:** The hematological parameters in male and female animals ^$^.

Treatment #	WBC	Lymph	Mon	Gran	Lymph%	Mon%	Gran%	RBC	HGB	HCT	MCV	MCH	MCHC	RDW	PLT	MPV	PDW
	10^9^/L	10^9^/L	10^9^/L	10^9^/L	%	%	%	10^12^/L	g/L	%	fL	pg	g/L	%	10^9^/L	fL	
Female																	
NC	4.80 ± 0.99	3.90 ± 0.78	0.12 ± 0.04	0.78 ± 0.21	81.05 ± 2.35	2.65 ± 0.84	16.30 ± 1.70	10.52 ± 0.36	147.00 ± 7.90	56.22 ± 2.14	53.70 ± 1.20	13.68 ± 0.46	255.83 ± 14.25	15.95 ± 0.49	1123.33 ± 333.01	6.70 ± 0.54	17.13 ± 0.27
H1	6.28 ± 0.47	5.08 ± 0.28	0.13 ± 0.05	1.07 ± 0.29	80.68 ± 4.20	2.55 ± 0.72	16.77 ± 3.71	10.30 ± 0.38	141.33 ± 5.47	54.02 ± 2.85	52.48 ± 0.94	13.70 ± 0.37	261.50 ± 9.77	15.40 ± 0.36	1188.33 ± 291.95	6.77 ± 0.30	17.32 ± 0.18
H2	6.45 ± 2.33	5.43 ± 2.09	0.17 ± 0.05	0.85 ± 0.26	83.90 ± 3.13	2.68 ± 0.84	13.42 ± 2.41	10.56 ± 0.34	146.67 ± 6.53	54.80 ± 2.10	52.00 ± 1.15	13.85 ± 0.55	267.33 ± 13.82	16.07 ± 0.74	1284.83 ± 416.37	6.53 ± 0.69	17.02 ± 0.25
M1	6.48 ± 2.42	5.37 ± 2.17	0.20 ± 0.09	0.92 ± 0.27	81.97 ± 3.59	3.18 ± 1.00	14.85 ± 3.09	9.70 ± 0.32	140.50 ± 3.21	53.42 ± 1.86	55.12 ± 0.80	14.43 ± 0.20	263.00 ± 6.20	15.88 ± 0.75	1251.33 ± 279.63	6.88 ± 0.47	17.32 ± 0.21
M2	6.68 ± 2.17	5.37 ± 1.80	0.13 ± 0.08	1.18 ± 0.32	79.85 ± 1.93	2.53 ± 0.56	17.62 ± 1.90	9.98 ± 0.46	143.00 ± 6.16	55.38 ± 2.39	55.53 ± 0.39	14.28 ± 0.26	257.67 ± 3.88	15.25 ± 0.29	1241.17 ± 253.09	6.95 ± 0.46	17.30 ± 0.33
L1	6.43 ± 2.73	5.30 ± 2.36	0.15 ± 0.08	0.98 ± 0.34	81.32 ± 3.21	2.63 ± 0.81	16.05 ± 2.55	10.40 ± 0.76	140.17 ± 10.26	54.77 ± 4.38	52.67 ± 0.56	13.42 ± 0.17	255.67 ± 3.50	16.20 ± 0.41	1274.00 ± 178.81	6.62 ± 0.55	17.00 ± 0.20
L2	6.15 ± 2.21	5.12 ± 2.09	0.13 ± 0.05	0.90 ± 0.17	81.33 ± 4.52	2.87 ± 0.74	15.80 ± 3.90	9.73 ± 0.17	137.67 ± 5.92	52.12 ± 1.16	53.63 ± 1.73	14.27 ± 0.48	266.83 ± 6.68	16.00 ± 0.68	1239.33 ± 347.23	6.65 ± 0.32	17.32 ± 0.33
Male																	
NC	7.38 ± 2.68	5.87 ± 2.29	0.18 ± 0.08	1.33 ± 0.36	78.80 ± 2.96	2.48 ± 0.36	18.72 ± 2.70	10.38 ± 0.79	139.50 ± 12.86	54.00 ± 3.34	52.17 ± 2.41	13.38 ± 0.29	257.50 ± 13.43	17.32 ± 2.55	1327.50 ± 239.63	6.57 ± 0.27	17.07 ± 0.43
H1	9.88 ± 2.47	8.13 ± 2.06	0.18 ± 0.10	1.57 ± 0.45	81.87 ± 3.06	2.18 ± 0.47	15.95 ± 2.77	9.86 ± 0.90	137.00 ± 13.81	52.25 ± 3.95	53.15 ± 2.27	13.85 ± 0.27	261.50 ± 15.42	17.27 ± 1.29	1421.67 ± 437.78	6.63 ± 0.36	17.37 ± 0.36
H2	9.83 ± 1.93	7.97 ± 1.71	0.18 ± 0.04	1.68 ± 0.26	80.50 ± 2.30	2.05 ± 0.31	17.45 ± 2.03	10.22 ± 0.40	143.17 ± 4.54	51.27 ± 3.07	50.17 ± 1.31	13.90 ± 0.27	279.50 ± 11.91	19.82 ± 1.71	1299.83 ± 227.69	6.37 ± 0.30	16.87 ± 0.45
M1	10.17 ± 1.52	8.74 ± 1.03	0.37 ± 0.14	1.95 ± 0.46	79.85 ± 2.99	3.15 ± 0.96	17.00 ± 2.21	10.07 ± 0.57	136.00 ± 7.51	52.48 ± 3.91	52.18 ± 2.15	13.45 ± 0.16	259.17 ± 13.45	17.03 ± 1.88	1546.75 ± 167.92	6.52 ± 0.31	17.23 ± 0.45
M2	9.70 ± 3.68	7.75 ± 3.17	0.25 ± 0.10	1.70 ± 0.52	79.23 ± 3.84	2.57 ± 0.64	18.20 ± 3.35	10.26 ± 0.69	142.17 ± 11.32	51.33 ± 5.09	50.05 ± 2.42	13.82 ± 0.26	277.33 ± 11.38	19.65 ± 2.26	1485.00 ± 176.13	6.67 ± 0.34	16.92 ± 0.56
L1	8.77 ± 2.72	7.17 ± 2.16	0.17 ± 0.08	1.43 ± 0.52	82.12 ± 2.41	1.88 ± 0.45	16.00 ± 2.28	10.33 ± 0.53	141.50 ± 5.68	50.03 ± 4.34	48.43 ± 2.37	13.67 ± 0.27	275.25 ± 17.25	19.53 ± 2.33	1328.17 ± 317.71	6.38 ± 0.37	16.43 ± 0.55
Ref	0.8–10.6	0.6–8.9	0.04–1.4	0.23–3.6	40–92	0.9–18	6.5–50	6.5–11.5	110–165	35–55	41–55	13–18	250–360	12–20	400–1600	4.0–6.2	12.0–17.5

^$^ Abbreviations: WBC, white blood cell count; Lymph, lymphocyte count; Mon, monocyte count; Gran, granulocyte count; Lymph%, lymphocyte percentage; Mon%, monocyte percentage; Gran%, granulocyte percentage; RBC, red blood cell count; HGB, hemoglobin; HCT, hematocrit; MCV, mean corpuscular volume; MCH, mean corpuscular hemoglobin; MCHC, mean corpuscular hemoglobin concentration; RDW, red cell distribution width; PLT, platelet count; MPV, mean platelet volume; PDW, platelet distribution width; Ref, reference value range. # NC: negative control, H1: high-dose *L. xujianguonis* HT111-2 gavage, H2: high-dose *L. xujianguonis* 06-2 gavage, M1: middle-dose *L. xujianguonis* HT111-2 gavage, M2: middle-dose *L. xujianguonis* 06-2 gavage, L1: low-dose *L. xujianguonis* HT111-2 gavage, L2: low-dose *L. xujianguonis* 06-2 gavage. Number of mice per group = 6 for each gender.

**Table 4 nutrients-17-03474-t004:** The biochemical parameters in male and female mice in the subacute toxicity study ^#^.

Treatment #	ALT	AST	BUN	CRE	TC	TG	CK	Na	K
Female									
NC	35.78 ± 3.30	160.68 ± 36.35	8.74 ± 0.49	13.00 ± 0.50	2.88 ± 0.08	0.44 ± 0.19	1235.53 ± 526.45	155.67 ± 0.81	7.09 ± 0.31
H1	34.40 ± 4.60	175.03 ± 42.05	8.46 ± 0.22	13.28 ± 0.86	2.75 ± 0.33	0.39 ± 0.07	1664.27 ± 636.88	157.70 ± 2.09	7.29 ± 0.80
H2	34.37 ± 3.13	170.40 ± 30.19	8.74 ± 0.35	12.80 ± 0.95	2.57 ± 0.11	0.22 ± 0.08	1443.42 ± 515.52	157.80 ± 1.11	7.08 ± 0.12
M1	31.32 ± 3.04	173.12 ± 36.41	8.16 ± 0.71	13.10 ± 0.78	2.59 ± 0.35	0.30 ± 0.12	1707.08 ± 463.81	157.75 ± 0.42	6.66 ± 0.19
M2	30.02 ± 2.26	146.05 ± 20.65	8.18 ± 0.92	12.60 ± 0.81	2.57 ± 0.12	0.27 ± 0.07	1342.58 ± 388.33	156.60 ± 1.70	6.79 ± 0.40
L1	34.95 ± 3.67	229.67 ± 88.22	7.60 ± 0.68	13.22 ± 0.29	2.51 ± 0.16	0.24 ± 0.08	2025.22 ± 480.65	155.22 ± 0.84	6.54 ± 0.25
L2	32.52 ± 4.36	192.38 ± 64.60	7.60 ± 0.45	12.72 ± 1.06	2.60 ± 0.15	0.26 ± 0.11	1754.78 ± 600.88	155.16 ± 1.74	6.59 ± 0.46
Male									
NC	40.95 ± 5.83	169.57 ± 25.22	11.05 ± 0.82	11.90 ± 0.94	3.00 ± 0.21	0.53 ± 0.10	1745.62 ± 616.85	158.73 ± 1.87	7.90 ± 0.48
H1	36.40 ± 4.63	140.02 ± 30.40	10.08 ± 0.59	12.80 ± 0.54	3.01 ± 0.22	0.61 ± 0.08	1242.08 ± 469.88	156.60 ± 1.17	7.73 ± 0.63
H2	33.77 ± 4.18	165.02 ± 38.64	10.94 ± 1.18	13.02 ± 1.38	2.99 ± 0.13	0.51 ± 0.16	1698.65 ± 535.90	157.87 ± 1.10	7.16 ± 0.25
M1	37.82 ± 4.60	159.08 ± 25.02	8.95 ± 1.02	12.57 ± 0.56	3.03 ± 0.15	0.68 ± 0.30	1509.68 ± 413.40	156.88 ± 1.07	7.30 ± 0.81
M2	34.67 ± 3.03	156.87 ± 18.80	9.95 ± 1.12	12.48 ± 0.92	2.95 ± 0.18	0.53 ± 0.18	1735.52 ± 487.55	156.78 ± 1.25	7.19 ± 0.31
L1	34.96 ± 4.91	133.38 ± 22.16	9.95 ± 1.81	13.52 ± 4.25	3.15 ± 0.13	0.47 ± 0.07	1666.77 ± 529.52	156.45 ± 0.24	6.87 ± 0.37
L2	35.00 ± 9.28	168.70 ± 35.61	11.62 ± 2.47	15.04 ± 3.99	2.77 ± 0.35	0.42 ± 0.18	2096.00 ± 809.74	156.90 ± 1.27	7.19 ± 0.73

^#^ Abbreviations. ALT: alanine aminotransferase, AST: aspartate aminotransferase, BUN: blood urea nitrogen, CRE: creatinine, TC: total cholesterol, TG: triglycerides, CK: creatine kinase, Na: sodium, K: potassium. # NC: negative control, H1: high-dose *L. xujianguonis* HT111-2 gavage, H2: high-dose *L. xujianguonis* 06-2 gavage, M1: middle-dose *L. xujianguonis* HT111-2 gavage, M2: middle-dose *L. xujianguonis* 06-2 gavage, L1: low-dose *L. xujianguonis* HT111-2 gavage, L2: low-dose *L. xujianguonis* 06-2 gavage. Number of mice per group = 6 for each gender.

## Data Availability

The original contributions presented in this study are included in the article/Appendix A. Further inquiries can be directed to the corresponding authors.

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
