# Peer review of "Safety Assessment and Probiotic Potential of a Novel Species *Lactobacillus xujianguonis"

_nutrients, 2025, doi:10.3390/nu17213474_

Round 1
Reviewer 1 Report
Comments and Suggestions for Authors
Dear Authors,
Your manuscript entitled “Safety assessment and probiotic potential of a novel species Lactobacillus xujianguonis” presents a well-conducted and relevant study addressing the safety and functional potential of a newly identified Lactobacillus strain. The topic is timely, as the characterization of new probiotic candidates is an important step toward the development of safe and effective functional foods.
Overall, the paper is scientifically solid, well written, and clearly presented. The experimental design follows recognized international standards (OECD, FAO/WHO, CLSI), and the results are consistent and well interpreted. Figures and tables are clear, and the conclusions are supported by the data. Only minor revisions are needed to further improve clarity, methodological transparency, and linguistic flow.
-
Methodological detail:
Please indicate the exact number of animals per group and whether both sexes were included in all analyses. Clarify if randomization and/or blinding were applied during the in vivo experiments. -
Genomic aspects:
Since you highlight the absence of pathogenic traits, it would strengthen the paper to mention (if available) genomic screening results for antibiotic resistance or virulence-related genes. If not available, a short comment acknowledging this as a limitation would be appropriate. -
Discussion and limitations:
The discussion is clear but could be slightly expanded to acknowledge the limitations of the study (e.g., lack of long-term exposure or microbiota analysis) and outline potential directions for future research. -
Language and readability:
The English is good and fully understandable, but a light linguistic polishing by a fluent or native speaker is recommended to improve flow, avoid repetitions (e.g., “the strain”, “this study”), and refine sentence structure.
Reviewer 2 Report
Comments and Suggestions for Authors
This manuscript investigates the probiotic potential and preclinical safety of a novel Lactobacillus species, Lactobacillus xujianguonis, using two strains. The authors present comprehensive in vitro and in vivo analyses, including gastrointestinal tolerance, autoaggregation, antimicrobial activity, antibiotic susceptibility, and acute and 28-day repeated oral toxicity in mice. The work addresses an important need for strain-specific safety evaluation of novel probiotics, which is highly relevant for food and therapeutic applications. However, the manuscript requires revision to clarify methodological details, improve data presentation, and better contextualize the findings.
Major Comments:
- The probiotic characterization relies on acid/bile tolerance, autoaggregation, and hydrophobicity as proxies for intestinal colonization, but no direct adhesion or colonization assays in intestinal epithelial cells or in vivo are provided. Including such data would substantiate the claims of gastrointestinal persistence.
- Antimicrobial activity results are described qualitatively. Including quantitative data (e.g., inhibition zone diameters with statistical comparisons) would improve the robustness and interpretability of these findings.
- The in vivo studies assess acute and 28-day toxicity in healthy mice, but the translational relevance for humans, especially immunocompromised or microbiota-disrupted hosts, is not addressed. Additional discussion or data on susceptibility in vulnerable models would strengthen the safety claims.
- RT-qPCR analysis of gut barrier genes shows minimal changes, but no protein-level validation (e.g., Western blot or immunohistochemistry) is provided. Gene expression alone is insufficient to confirm functional integrity of the epithelial barrier. Also, the methods describing RT-qPCR analysis lack information on normalization strategies and validation of reference genes; clarifying this would strengthen confidence in the transcriptional data.
- The manuscript could benefit from a clearer discussion of limitations, including long-term safety, studies in immunocompromised hosts, and potential horizontal gene transfer of antibiotic resistance genes.
Minor Comments:
- Several typographical errors and inconsistent use of strain names (e.g., HT111-2 vs HT1112) should be corrected for clarity.
- The references to prior probiotic studies could be updated to include more recent findings on safety assessments of novel Lactobacillus species.
- In the introduction, the rationale for choosing Marmota himalayana as a source for isolation could be briefly justified to highlight novelty.
Reviewer 3 Report
Comments and Suggestions for Authors
The authors present an interesting and timely study entitled “Safety assessment and probiotic potential of a novel species Lactobacillus xujianguonis”, addressing an important topic in probiotic safety evaluation. The manuscript demonstrates commendable effort in combining in vitro functional assays with in vivo toxicological studies, providing valuable preliminary insights into the potential of this novel species. However, several aspects of the work—including methodological detail, interpretation of results, contextualization within existing literature, and clarity of presentation—require substantial refinement. A major revision is therefore necessary to strengthen the scientific rigor, ensure compliance with international standards, and improve the overall readability of the current version.
ABSTRACT
# Lines 16–18 (Background):
- The background is concise but too general. It mentions “emerging evidence” challenging lactobacilli safety but does not specify what evidence (e.g., bacteremia in immunocompromised patients, antibiotic resistance concerns). Without context, the rationale feels underdeveloped.
- The novelty of L. xujianguonis is stated, but its origin (isolated from Marmota himalayana) is not mentioned, which would strengthen the justification.
# Lines 19–23 (Objective and Methods):
- The objective is clearly stated, but the phrase “industrial applicability” is too broad and not directly supported by the methods described.
- The methods are listed in a long sentence that is dense and difficult to follow. Key details (e.g., OECD guidelines for toxicity studies, number of animals, sex distribution) are missing.
- The abstract does not mention genomic safety assessment, which is increasingly required for probiotic safety evaluation.
# Lines 24–32 (Results):
- The results are presented in a structured way, but the language is overstated (“robust probiotic traits,” “confirmed susceptibility”). Abstracts should avoid definitive claims based solely on in vitro and short-term animal studies.
- The reporting of results is imbalanced: too much detail on specific markers (TNF-α, IL-6, IFN-γ, Occludin, Zo1, Muc2) without context, while other important findings (e.g., absence of adverse clinical signs, body weight stability) are not emphasized.
- The phrase “Histology found no bacterial translocation” (line 32) is vague—was this based on culture, staining, or molecular detection?
# Lines 33–34 (Conclusion):
- The conclusion is overstated: claiming “evidence for food-grade safety and industrial applicability” and “supporting advancement to clinical trials” is premature. Regulatory frameworks require genomic analysis, long-term studies, and human data before such claims.
- The conclusion should be more cautious, framing the findings as preliminary evidence that supports further investigation.
# Line 36 (Keywords):
- Keywords are relevant, but “acute toxicity” and “subacute toxicity” are too generic. More specific terms (e.g., “oral toxicity,” “murine model,” “antibiotic susceptibility”) would improve discoverability.
INTRODUCTION
# Lines 38–46:
- The introduction provides a standard definition of probiotics and outlines their therapeutic potential. However, it is overly generic and does not sufficiently highlight the knowledge gap that justifies this study.
- The mechanisms of action are briefly mentioned but not critically linked to the rationale for evaluating a novel species.
# Lines 47–53:
- The discussion of safety is appropriate, but it lacks depth and nuance. For example, while Lactobacillus bacteremia is mentioned, the incidence rates, clinical contexts, and risk-benefit considerations are not elaborated.
- The phrase “Lactobacillus bacteriaemia has been associated with their probiotic use” (line 51) is too strong without specifying that such cases are rare and often linked to immunocompromised patients.
# Lines 54–62:
- The reference to WHO and FDA guidelines is important, but the description is incomplete. The authors should also mention EFSA (European Food Safety Authority) and its Qualified Presumption of Safety (QPS) framework, which is highly relevant for probiotic evaluation.
- The sentence “must be undergo individual assessment” (line 54–55) contains a grammatical error and should be corrected.
# Lines 63–75:
- The isolation of L. xujianguonis from Marmota himalayana is interesting, but the relevance of a non-human source is not critically discussed. Why is this species a promising candidate despite being isolated from a wild animal host?
- The phylogenetic clustering with L. johnsonii and L. acidophilus is mentioned, but the authors should explain why this relationship increases its probiotic potential.
- The study aim is clearly stated, but the description of methods (acid resistance, aggregation, pathogen inhibition, antibiotic susceptibility, hemolysis, gelatinase, in vivo toxicity) is too detailed for an Introduction. This belongs in the Methods section.
METHODS
# Lines 77–91:
- The strain information is provided, but the strain nomenclature is inconsistent (e.g., “CGMCC NO.1.13855” vs. “CGMCC NO.23437” – formatting should be standardized).
- The description of S. flexneri storage (line 84–85) is unclear and contains a typographical error (“tconserved”).
- Culture conditions are described, but oxygen requirements (strict anaerobic vs. facultative) are not justified. Why was sheep blood supplementation used for L. xujianguonis?
# Lines 93–102:
- The survival rate formula (line 100) is unclear and possibly incorrect. The use of “lg” suggests logarithmic values, but the formula as written is ambiguous.
- No mention of biological replicates or statistical methods for survival rate calculation.
# Lines 103–113:
- The methods are adapted from Zulkhairi et al., but critical details are missing: incubation atmosphere (aerobic/anaerobic), number of replicates, and whether controls were included.
- The 24 h equilibration with xylene (line 109–110) is unusually long; most protocols use shorter times (e.g., 10–30 min). This deviation should be justified.
# Lines 114–125:
- The Oxford cup method is described, but quantification criteria (e.g., how inhibition zones were measured, replicates, statistical analysis) are not specified.
- The choice of pathogens is appropriate, but the rationale for including both EHEC and EAEC is not explained.
# Lines 126–132:
- The use of CLSI M45 is correct, but the E-test method requires more detail: inoculum density, incubation conditions, interpretation criteria.
# Lines 143–151:
- Ethical approval is mentioned, which is good. However, animal randomization and blinding procedures are not described.
- Housing conditions are provided, but diet composition is not specified, which could influence gut microbiota outcomes.
# Lines 152–161:
- The OECD guideline is cited, but modifications (line 153–154) are not clearly explained.
- Only female mice were used; justification for excluding males is missing.
- Sample size (n=6 per group) is small and may lack statistical power.
# Lines 162–176:
- The study design is comprehensive, but group allocation is confusing: seven groups are mentioned (line 165), but only six are described (control + 5 treatment groups). Clarification is needed.
- The NOAEL determination is mentioned, but criteria for defining adverse effects are not specified.
- RT-qPCR analysis of colon tissues is included, but primer sequences, reference genes, and normalization methods are not described.
# Lines 177–186:
- Instruments are listed, but parameters measured (e.g., ALT, AST, creatinine, cytokines) are not specified.
- No mention of quality control procedures or calibration of instruments.
# Lines 187–191:
- The fixation and staining methods are standard, but criteria for histopathological scoring are not described.
- Only six organs were examined; justification for excluding others (e.g., brain, reproductive organs) is missing.
# Lines 192–201:
- The RNA extraction and cDNA synthesis workflow is described, but critical details are missing:
- RNA quality control (e.g., A260/280 ratios, RIN values) is not reported.
- cDNA synthesis conditions (reaction volume, input RNA amount) are not specified.
- RT‑qPCR cycling conditions (denaturation, annealing, extension temperatures/times, number of cycles) are absent.
- No mention of biological replicates (number of mice per group analyzed) or technical replicates (triplicates per sample).
# Lines 202–206:
- The rationale for using Spearman correlation to assess “repeatability of samples and quality control samples” (line 204–205) is unclear and not standard practice. Repeatability is usually assessed by coefficient of variation (CV) or intraclass correlation coefficients (ICC).
- No mention of how normality and homogeneity of variance were tested before applying ANOVA.
DISCUSSION
# Lines 322–330:
- The authors provide a general background on Lactobacillus safety and EFSA’s QPS framework. However, the knowledge gap is not sharply defined. The novelty of L. xujianguonis is acknowledged, but the Discussion does not critically compare its properties with established probiotic strains.
- The section reiterates the study aim rather than synthesizing findings in the context of existing literature.
# Lines 331–342:
- The discussion of adhesion, acid/bile tolerance, and antibacterial activity is appropriate, but the interpretation is overstated (“exceptional acid-bile tolerance,” line 340). The data should be contextualized against benchmarks from well-characterized probiotics (e.g., L. rhamnosus GG).
- The link between auto-aggregation, hydrophobicity, and colonization is mentioned, but the authors do not discuss limitations of in vitro assays as predictors of in vivo colonization.
# Lines 343–347:
- The finding of full susceptibility is important, but the authors do not address whether genomic analysis was performed to detect transferable resistance genes. This is a critical omission for safety assessment.
# Lines 348–364:
- The acute toxicity results are described in detail, but the Discussion repeats methods and results rather than interpreting them.
- The claim that LD50 exceeds tested concentrations (line 358–359) is speculative, as LD50 was not directly determined.
# Lines 365–372:
- The authors correctly highlight the importance of subacute testing, but again, the section is descriptive rather than analytical. No discussion of potential subtle effects (e.g., microbiota shifts, immune modulation) is provided.
# Lines 373–387:
- The discussion is overly detailed and reads like a results section. Instead of listing parameters, the authors should interpret whether these findings align with or differ from prior probiotic safety studies.
- The phrase “metabolic panel analysis (TC, TG) and cardiac biomarker screening (CK, Na, K) collectively revealed unperturbed cardiometabolic homeostasis” (lines 385–386) is overly broad and not critically evaluated.
# Lines 388–395:
- The authors conclude absence of immunostimulation, but they do not discuss the limitations of cytokine profiling (e.g., short observation window, lack of challenge model).
- The sentence “Chances of immune factors… confirms immunomodulatory properties” (line 391) is grammatically incorrect and unclear.
# Lines 396–403:
- The preservation of Occludin, Zo1, and Muc2 expression is interesting, but the authors should acknowledge that mRNA expression does not always correlate with protein function.
- No discussion of whether these markers were validated at the protein level (e.g., Western blot, immunohistochemistry).
# Lines 404–411:
- The extrapolation of NOAEL to a human equivalent dose (HED) is problematic. The calculation method is not explained, and the claim of a “192-fold safety margin” (line 409–410) is overstated without considering interspecies differences, microbiota interactions, or long-term exposure.
- The conclusion reads more like a final summary than a critical discussion.
CONCLUSIONS
# Lines 412–414 (Beneficial potential):
- The conclusion restates results (acid/bile tolerance, aggregation, antibacterial activity) but does not critically evaluate their biological significance. For example, how do these traits compare quantitatively to benchmark probiotic strains such as L. rhamnosus GG or L. plantarum?
- The term “beneficial potential” is vague and should be more specific (e.g., “in vitro probiotic-associated traits”).
# Lines 415–417 (Safety profile):
- The conclusion claims “acceptable safety profiles” based on nine antibiotics, hemolysis/gelatinase absence, and short-term toxicity studies. However, this is overstated:
- Only nine antibiotics were tested; EFSA guidelines recommend a broader panel and genomic screening for transferable resistance genes.
- Safety cannot be confirmed solely by absence of hemolysis/gelatinase; other virulence factors should be excluded.
- Acute and 28-day studies are insufficient to establish long-term safety.
# Lines 417–418 (Basis for clinical trials):
- The statement that findings “provide the basis for further clinical trials” is premature. Regulatory frameworks (EFSA, FDA) require genomic characterization, toxicological validation, and human pilot studies before clinical trials.
# Lines 419–421 (Future studies):
- The authors acknowledge the need for long-term studies, cross-species models, and immunocompromised hosts, which is appropriate. However, they omit other critical next steps: such as, whole-genome sequencing to exclude transferable resistance/virulence genes, functional validation of intestinal barrier effects at the protein level, and microbiota interaction studies to assess ecological safety.
Comments on the Quality of English LanguageThe manuscript is generally understandable and conveys the intended scientific content; however, the quality of English requires improvement to meet the standards of the journal. Several issues were noted:
- Grammar and syntax: There are occasional grammatical errors (e.g., “must be undergo” in the Introduction; “was served as quality control” in Methods) and awkward phrasing that reduce clarity.
- Word choice and tone: Some expressions are overstated or imprecise (e.g., “exceptional tolerance,” “confirmed biosafety”), which should be replaced with more cautious and objective scientific language.
- Consistency: Terminology and formatting are sometimes inconsistent (e.g., strain names, units, abbreviations such as CFU/kg, gene names, and P‑values). These should follow international conventions.
- Clarity and conciseness: Several sentences are overly long and descriptive, particularly in the Discussion, where results are repeated rather than synthesized. Shorter, more precise sentences would improve readability.
- Typographical errors: Minor typographical mistakes (e.g., “tconserved,” “comparable to the normal control”) should be corrected.
Round 2
Reviewer 3 Report
Comments and Suggestions for Authors
The manuscript entitled “Safety assessment and probiotic potential of a novel species Lactobacillus xujianguonis” has been carefully evaluated. After the first round of review, the authors have thoroughly addressed the comments and suggestions provided by the reviewers. The revisions have improved the clarity, methodological detail, and overall presentation of the study. The manuscript is now well-structured, the data are clearly presented, and the conclusions are well supported by the results. The study makes a valuable contribution to the field of probiotic research and microbial safety assessment. I recommend acceptance of the manuscript in its current version.